# Towards Human-Level Bimanual Dexterous Manipulation with Reinforcement Learning

**Yuanpei Chen**[1]      **Tianhao Wu**[2]      **Shengjie Wang**[1]      **Xidong Feng**[4]      **Jiechuan Jiang**[3]

**Stephen Marcus McAleer**[5]      **Hao Dong**[2]      **Zongqing Lu**[3,1]      **Song-Chun Zhu**[1,6]

**Yaodong Yang**[1,6,†]

## Abstract

Achieving human-level dexterity is an important open problem in robotics. However, tasks of dexterous hand manipulation, even at the baby level, are challenging to solve through reinforcement learning (RL). The difficulty lies in the high degrees of freedom and the required cooperation among heterogeneous agents (e.g., joints of fingers). In this study, we propose the **Bi**manual **Dex**terous **Hands** Benchmark (Bi-DexHands), a simulator that involves two dexterous hands with tens of bimanual manipulation tasks and thousands of target objects. Specifically, tasks in Bi-DexHands are designed to match different levels of human motor skills according to cognitive science literature. We built Bi-DexHands in the Issac Gym; this enables highly efficient RL training, reaching 30,000+ FPS by only one single NVIDIA RTX 3090. We provide a comprehensive benchmark for popular RL algorithms under different settings; this includes Single-agent/Multi-agent RL, Offline RL, Multi-task RL, and Meta RL. Our results show that the PPO type of on-policy algorithms can master simple manipulation tasks that are equivalent up to 48-month human babies (e.g., catching a flying object, opening a bottle), while multi-agent RL can further help to master manipulations that require skilled bimanual cooperation (e.g., lifting a pot, stacking blocks). Despite the success on each single task, when it comes to acquiring multiple manipulation skills, existing RL algorithms fail to work in most of the multi-task and the few-shot learning settings, which calls for more substantial development from the RL community. Our project is open sourced at https://github.com/PKU-MARL/DexterousHands.

## 1   Introduction

Humans have a skillful ability to manipulate objects of different shapes, sizes, and materials, which rely on the dexterity of our hands and fingers. Building a robot inspired by human hands that can autonomously manipulate various objects has always been an important component of the robotics field [1]. The development of human dexterity begins in infancy and is influenced by what the physical environment provides, including the objects available to the child [2]. As infants and children develop

---

[1]Institute for AI, Peking University
[2]Center on Frontiers of Computing Studies, Peking University
[3]School of Computer Science, Peking University
[4]University College London
[5]Carnegie Mellon University
[6]Beijing Institute for General Artificial Intelligence
†: Corresponding to <yaodong.yang@pku.edu.cn>

36th Conference on Neural Information Processing Systems (NeurIPS 2022) Track on Datasets and Benchmarks.

physical and intelligence, they are more likely to attempt complex movements, and often learn dexterity through attempting movements and the consequences of their actions [3, 4, 5]. Similarly, robot dexterity can not be a constant program pre-set in the laboratory. To acquire the capability of object manipulations in the real world, robots must be able to learn dexterous manipulation skills as if we were infants. As a result, we expect robots to learn to master the ability of dexterous manipulation at the human level from daily tasks.

Recently, reinforcement learning (RL) algorithms have outperformed human experts in many fields that require decision makings [6, 7]. In contrast to the traditional control methods, RL can complete some challenging tasks in learning dexterous in-hand manipulation [8, 9, 10] or grasping [11, 12]. However, manipulation that generates changes on the object is still difficult [13]. More difficult is generalization across tasks, although previous work can achieve simple level of tasks such as throwing [14], sliding [15], poking [16], pivoting [17], and pushing [18], but is still difficult to perform well in unstructured or contact-rich environments, which require the ability to combine and generalize complex manipulation skills. In a nutshell, reaching human-level sophistication of hand dexterity and bimanual coordination remains an open challenge for modern robotics researchers.

To help solve the problems mentioned above and let robots have the same dexterous manipulation ability as humans, we developed a novel benchmark on bimanual dexterous manipulation for RL algorithms along with a diverse set of tasks and objects named **Bi-DexHands**. We follow the principle of Fine Motor Subtest (FMS) [19] to design tens of tasks, which provides the opportunities to observe and evaluate specific skills that demonstrate a child's ability to use their hands to play with toys, manipulate objects, and use tools. Next, we tested the baselines of various model-free RL algorithms to show the ability of the baseline algorithm in these tasks, not only the standard RL algorithms but also multi-agent RL (MARL), offline RL, multi-task RL, and Meta RL algorithms, each of them focuses on the bimanual collaboration, learning from demonstration, and task generalization, respectively. Our major goal is to facilitate researchers to master human-level bimanual dexterous manipulations with RL. Not limited to this, we also hope this study to provide a new platform for the community of RL, robotics, and cognitive science. Bi-DexHands are developed with the following key features:

- **Isaac Gym Efficiency**: Building on the Isaac Gym [20] simulator, Bi-DexHands supports running thousands of environments simultaneously. On one NVIDIA RTX 3090 GPU, Bi-DexHands can reach 30,000+ mean FPS by running 2,048 environments in parallel.

- **Comprehensive RL Benchmark**: We provide the first bimanual manipulation task environment for common RL, MARL, offline RL, multi-task RL, and Meta RL practitioners, along with a comprehensive benchmark for SOTA continuous control model-free RL methods.

- **Heterogeneous-agent Cooperation**: Agents in Bi-DexHands (*i.e.*, joints, fingers, hands,...) are genuinely heterogeneous; this is different from common multi-agent environments such as SMAC [21] where agents can simply share parameters to solve the task.

- **Task Generalization**: We introduced a variety of dexterous manipulation tasks (e.g., hand over, lift up, throw, place, put...) as well as enormous target objects from the YCB [22] and SAPIEN [23] dataset, thus allowing meta-RL and multi-task RL algorithms to be tested on the task generalization front.

- **Cognition**: We provided the underlying relationship between our dexterous tasks and the motor skills of humans at different ages. This facilitates researchers on studying robot skill learning and development, in particular in comparison to humans.

## 2 Related Work

Today, robots are skilled in some repetitive and familiar environments like assembled in the factory. Grasping is a milestone in robotics manipulation. For decades, researchers have been working to establish a stable grasping theory [24]. However, most previous methods have relied on various assumptions, such as known object information or no uncertainty in the process. In recent years, data-driven approaches have been successful in this regard, being able to deal with uncertainty in perception and generate grasping methods for known, familiar, and even unknown objects in real-time [25]. Grasping is only a part of the manipulation. Today's robots can perform some simple behaviors like grasping, pushing, and throwing. But it is still difficult to manipulate in unstructured scenes and contact-rich situations. Moving objects while in-hand manipulation is also a complex

challenge. One step to address this challenge is to use hands with intrinsic dexterity [26, 27], which often mimic human hands [28]. Another undeveloped area is bimanual manipulation, a method of using a second hand to provide additional dexterity [29, 30]. Learning for manipulation is important for robots to continuously learn and achieve intelligent control. It is especially suitable for modeling manipulation on complex non-rigid objects and reducing control dimensions [31], but it still suffers from problems such as lack of accurate models, reality gaps, and difficulty in collecting expert data. There are many other robotic manipulation benchmarks [32, 33, 34], but none of them use dexterous hands. Therefore, our work proposes a bimanual dexterous manipulation benchmark, hoping to facilitate researchers to address the challenges of robotic manipulation we mentioned above.

Dexterous five-finger hands provide an essential tool to perform a multitude of tasks in human-centric environments. However, such dexterous manipulation remains a challenging problem because of the high dimensional actuation space and contact-rich model. Before the emergence of RL-based controllers, a large variety of manipulation tasks highly relied on accurate dynamics models and trajectory optimization methods [35, 36, 37]. For example, Williams et al. [38] used the model predictive path integral control (MPPI) method to perform the task successfully, dexterous manipulation of a cube. Charlesworth et al. [39] improved the MPPI method to make the handing over task between two hands tractable. Since RL simplifies the design process of the controller, model-agnostic approaches have become more and more popular in the field of robotic control [40, 41]. In terms of dexterous manipulation, many works achieve a significant improvement compared with traditional controllers. OpenAI et al. [9] developed an RL-based controller to reorient a block or a Rubik's cube. Considering the poor generalization of current approaches, Chen et al. [10] presented an efficient system for learning how to reorient a large number of objects without access to shape information. Qin et al. [42, 43] perform learning from demonstrations for dexterous manipulation collected from teleoperation or video. While their studies demonstrate that RL enables efficient and scalable learning on single-hand manipulation, bimanual manipulation remains a hardship for model-free reinforcement learning [39]. In this paper, our benchmark provides a wide range of well-designed and challenging daily life scenarios for comprehensive RL algorithms, hoping to help the researcher toward master human-level bimanual dexterous manipulation.

## 3   Formulations & Algorithms

In order to create a platform toward mastering human-level dexterity, we use two Shadow Hands to manipulate in our environment. Shadow Hand [44] is a popular robotic hand usually used in some dexterous manipulation tasks. It is designed to resemble the typical human male hand in shape and size, and capable of performing a variety of flexible and delicate operations. Shadow Hand's DoF is shown in Fig.1, designed to mimic the human skeleton as much as possible. Concretely, the 24-DoF hand is actuated by 20 pairs of agonist-antagonist tendons, while the other four joints remain under-actuated.

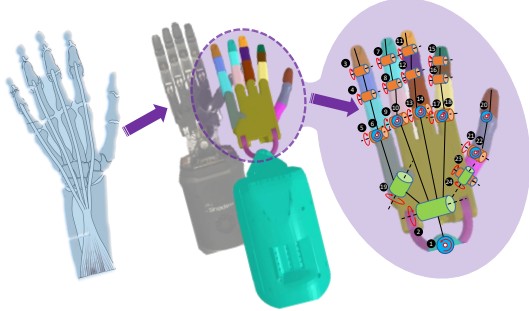

Figure 1: Degree-Of-Freedom (DOF) configuration of the Shadow Hand similar to the skeleton of a human hand.

Furthermore, our low-level controller runs at 1k Hz, as well as the RL-based policy outputs the relative positions of actuated joints at 30 Hz. It is worth noting that compared with previous studies, the base of the hand is not fixed in some tasks. Instead, the policy can control the position and orientation of the base within a restricted space, which takes advantage of the function of the wrist, thus making the Shadow Hand more bio-mimetic. Meanwhile, we can efficiently perform the task in real-world applications by linking the base to a robotic arm. Refer to Appendix A.1 for more details about the physical parameters of the Shadow Hand.

Our benchmark aims at providing solutions for bimanual dexterous manipulation in a comprehensive field of RL. To achieve that, We consider five RL formulations including: Single-agent RL, Multi-agent RL (MARL), Offline RL, Multi-task RL, and Meta-RL in Bi-DexHands. In the following part, we will introduce the detailed formulation and the corresponding implemented algorithms in our benchmark of these five RL formulations.

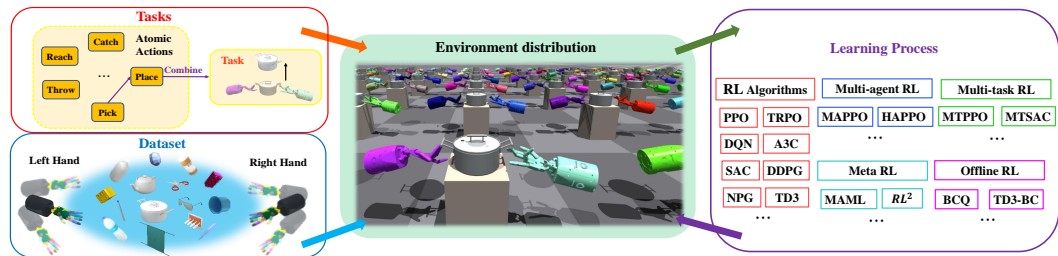

Figure 2: Framework of Bi-DexHands, a bechmark for learning bimanual dexterous manipulation.

**RL/MARL.** In order to evaluate the performance of RL/MARL [45, 46], we formulate our scenarios as a decentralized partially observable MDP (Dec-POMDP). The Dec-POMDP consists of 10 components, $Z = < \mathcal{N}, \mathcal{M}, S, \boldsymbol{O}, \boldsymbol{A}, \boldsymbol{\Pi}, P, R, \rho, \gamma >$. Initially, the robotic hands are manually separated as $\mathcal{N}$ agents, the set of which represents $\mathcal{M}$. When starting the simulation, the state of the environment (i.e., the information of robots and objects) is set at $s_0 \in S$ according to the initial distribution of states $\rho(s_0)$. Then at the time step $t$, $s_t$ represents the state, and the i-th agent receives an observation $o_t^i \in \boldsymbol{O}$ relying on $s_t$. Hereafter, the policy of the i-th agent, $\pi_i \in \boldsymbol{\Pi}$, takes the $o_t^i$ as input, and outputs an action $a_t^i \in A_i$. Additionally, we denote the joint action of all agents by $\boldsymbol{a_t} \in \boldsymbol{A}$, and the equation $\boldsymbol{A} = [A_1, ..A_{\mathcal{N}}]$ is naturally satisfied. After that, i-th agent can obtain a reward $r_t^i$ based on $R(s_t, \boldsymbol{a_t})$, as well as all agents transitions to the next state $s_{t+1}$ with the possibility of the transition function $P(s_{t+1}|s_t, \boldsymbol{a_t})$. The goal is to find the optimal policy $\boldsymbol{\Pi}$ to maximize the sum of rewards $\mathbb{E}_{\boldsymbol{\Pi}}[\sum_{t=0}^{T-1} \gamma^t \sum_{i=1}^{\mathcal{N}} r_t^i]$ in an episode with $T$ time steps. It should be pointed out that when $\mathcal{N} = 1$, it is the problem formulation of single-agent RL.

In this setting, We implemented state-of-the-art continuous single-agent RL algorithms, such as PPO [47], SAC [48], TRPO [49], DDPG [50], and TD3 [51] algorithms. Taking our continuous control and fully cooperative environments into consideration, we introduced HAPPO/HATRPO [52, 53, 54], MAPPO [55], IPPO [56], and MADDPG [57] algorithms.

**Offline RL.** Offline RL follows the formulation of standard MDP, where the goal is to maximize the expected return $\mathbb{E}_\pi[\sum_{t=0}^{T-1} \gamma^t r_t]$. However, in offline RL, the agent has to learn policy only using the transitions in previously collected dataset $\mathcal{D} = \{(s_t, a_t, s_{t+1}, r_t)\}$, without interacting with the environment. The fundamental challenge of offline RL is value errors of out-of-distribution actions [58]. We implemented BCQ [59], TD3+BC [58], and IQL [60] algorithms for offline RL.

**Multi-task RL.** Multi-task reinforcement learning aims to train a single policy $\pi(a|s, z)$, which can achieve good results on different tasks. $z$ represents an encoding of the task ID. The goal of our policy is to maximize the reward given by $\mathbb{E}_{\mathcal{T} \sim p(\mathcal{T})}[\mathbb{E}_\pi[\sum_{t=0}^{T-1} \gamma^t r_t]$, where $p(\mathcal{T})$ is a task distribution in our benchmark. In practice, multi-task RL adds the context vector corresponding to the type of environment (e.g., one-hot task ID) into states to learn a general skill. We implemented multi-task PPO, multi-task TRPO, and multi-task SAC algorithms for multi-task RL.

**Meta RL.** Meta RL [61], also known as "learning to learn", aims to gain the ability to train on tasks to extract the common features of these tasks, so as to quickly adapt to new and unseen tasks. In Meta-RL, both training and test environments are assumed to follow the same task distribution $p(\mathcal{T})$. In Bi-DexHands, we design some common structures between different tasks for meta-training to ensure that it can adapt efficiently to new tasks. Compared with Multi-task RL, Meta RL is not allowed to get direct task-level information such as task ID. It needs to solve entirely new tasks by task inference and adaptation purely based on interactions. We implemented model-agnostic meta learning (MAML) [62] and proximal meta-policy search (ProMP) [63] algorithms for Meta RL.

## 4 Bimanual dexterous manipulation benchmark

In this section, we will discuss the construction of Bi-DexHands, a benchmark for bimanual dexterous manipulation over diverse scenarios.

### 4.1 System design

As we mentioned before, the core of Bi-DexHands is to build up a learning framework for two Shadow Hands capable of diverse skills as humans, such as reaching, throwing, catching, picking

and placing. To be specific, Bi-DexHands consists of three components: datasets, tasks, and learning algorithms, as shown in Fig.2. Varying worlds provide a large number of basic settings for robots, including the configuration of robotic hands and objects. Meanwhile, a variety of tasks corresponding to children's behaviors at different ages make it possible to learn dexterity like a human. Combining a dataset and task, we can generate a specific environment or scenario for the following learning. Eventually, our experiments demonstrate that reinforcement learning is able to facilitate the robots to achieve some remarkable performance on such challenging tasks, and there is still some room for improvement and more difficult tasks for future work.

## 4.2 Construction of datasets

The construction of the datasets corresponds to the configuration of robots and objects. The core goal of datasets is to generate a large variety of scenes for robot learning. As we mentioned in the last part, the robots in our benchmark are two dexterous Shadow Hands. Other than the robots, the objects also play an essential role in constructing the datasets. For extending the types of tasks, we introduced a variety of objects from the YCB [22] and SAPIEN [23] datasets. Two datasets contain many everyday objects. Notably, the SAPIEN dataset provides many articulated objects with motion annotations and rendering material, which means these objects are close to the real ones significantly. Therefore, it provides a natural way to build a connection between the worlds of our benchmark and scenes of daily life. Concretely, Fig.2 shows the construction of datasets, and we can see that the object includes pots, pens, eggs, scissors, eyeglasses, doors, and other common tools. After defining the configuration of robots and the type of objects, we build the specific world based on the Isaac Gym simulator. Meanwhile, each world defines variable initial poses of robots and objects, providing a diverse set of environments.

## 4.3 Design of tasks

An infant's behavior experiences a multi-stage development, such as social, communication, and physical parts [64]. Particularly in bimanual dexterous manipulation, there are some relationships between some common behaviors of babies and the ages. To gain insights into the underlying relationships, we conducted an in-depth analysis and built a mapping between the baby's age and tasks according to the Fine Motor Subtest (FMS) [19]. As the baby's age increases, the difficulty of completing the designed tasks also increases, because the baby can complete more and more difficult behaviors as the body develops. So it is also of great importance to evaluate the performance of trained agents, because we can roughly point out agents' intelligence level by analogy with a baby's movement for bimanual dexterous manipulation. An overview of the correspondence of our tasks to the FMS is shown in Table.1. For more details on the tasks, please refer to Appendix A.2.

## 4.4 Design of Multi-task/Meta RL

The design of our Multi-task/Meta RL categories is generally similar to Meta-World [32], divided into ML1, MT1, ML4, MT4, ML20, and MT20. Each of our tasks has object variation, which as we can interact with different kinds of objects in daily life scenes, providing a foundation for us to learn dexterous manipulation like humans. In the following, we will introduce 6 tasks categories for Multi-task/Meta-RL. More details can refer to Appendix D.

**MT1&ML1: Learning a multi-task policy & Few-shot adaptation within one task:** Both ML1 and MT1 are categories for generalization ability within the same task, and their generalization ability is reflected in the ability to complete tasks under different goals. ML1 uses meta-reinforcement learning for few-shot adaptation, in which goal information will not be provided. MT1 uses the multi-task method for generalization, and the information on the goal will be provided in a fixed set.

**MT4&MT20: Learning a multi-task policy belonging to 4&20 training tasks:** MT4 and MT20 conduct policy training in 4&20 tasks and hope to complete all tasks in only one policy. In MT4, we hope to learn policy with similar human skills, so we try to combine similar tasks as much as possible. MT20 uses all of our 20 tasks. In MT4 and MT20, we use a one-hot task ID to represent different tasks, and the information on the goal will be provided in a fixed set.

**ML4&ML20: Learning a Few-shot adaptation for new 1&5 test tasks from 3&15 training tasks:** ML4 and ML20 are categories for learning meta-policies in 3&15 tasks respectively and hoping to adapt to new 1&5 testing tasks. There is no doubt that this is a difficult challenge. We choose the tasks which using the catch behavior for design in ML4. The ML20 requires adaptation in all 15 tasks with large differences designed according to baby intelligence, which is the most difficult

Table 1: Task name and the description of the human skill in the corresponding age. References under the human age are the cognitive science literature referenced for the behavior designed, and the difficulty level of the tasks is under the task name. Easy level tasks are more basic skills, medium level tasks need more precise control and finger dexterity, and hard level tasks require handing dynamic interaction and tool use.

| Human Task Name | Human's Skill Description | Age (months) | Demo |
|---|---|---|---|
| Push Block Easy | Child extends one or both arms forward and touches the block with any part of either hand | 5-6 [19, Chapter 3] |  |
| Open Scissor & Open Pen Cap Easy | They use one hand to hold a toy and the other hand manipulate it | 7 [65, Chapter 4] |  |
| Turn Button ON/OFF Easy | They can push and squish soft stuff or push hard things, like a button on a toy phone or popup toy | 11 [66, 11 months] |  |
| Swing Cup Easy | They can turn a ball on their toy mobile, a steering wheel on a toy car, or the faucet in the tub | 11 [66, 11 months] |  |
| Lift Pot & Lift Cup Easy | They can put a sippy cup to their mouth to drink | 12 [66, 12 months] |  |
| Door Open & Close Easy | Toddlers can open and close cupboards and oven doors | 13 [66, 13 months] |  |
| Re-Orientation Medium | Infant further refines this ability to differentiate individual finger movement and manipulate objects | 18 [65, Chapter 4] |  |
| Stack Block(2,6,8) Medium | Child stacks at least 2/6/8 blocks in any trial. | 2:22-28 6,8:33-42 [19, Chapter 3] |  |
| Pull a Ball into Bucket Medium | Child place s 10 pellets in the bottle in 60 seconds or less, one pellet at a time. | 22-28 [19, Chapter 3] |  |
| Open Bottle Cap (prismatic joint) Medium | Uses hands to twist things, like turning doorknobs or unscrewing lids | 30 [67, Table 6] |  |
| Catch Underarm Hard | Catches a large ball most of the time | 48 [67, Table 6] |  |
| Pour Water Hard | Serves himself food or pours water, with adult supervision | 48 [67, Table 6] |  |
| Two Catch Underarm Hard | Some adults can throw objects between two hands like magic | adult |  |

challenge in our benchmark. Similarly, we will variate the goal for each task, and will not provide task information, requiring Meta RL algorithms to identify the tasks.

# 5 Benchmarking reinforcement learning algorithms

In this section, we conduct a full benchmark of the RL algorithms in Bi-DexHands. We firstly quantify our environment speed to demonstrate the running efficiency of Bi-DexHands. Then We offer the benchmark results and corresponding discussion and analysis on those five RL formulations. All of our experiments are run with Intel i7-9700K CPU @ 3.60GHz and NVIDIA RTX 3090 GPU. For the hyperparameters of all algorithms, please refer to Appendix B.

## 5.1 Environmental speed

Thanks to Isaac Gym's high-performance GPU parallel simulating capabilities, we can greatly improve the sampling efficiency of our RL algorithm while using fewer computing resources. We believe that the high sampling efficiency improves the exploration ability of the RL algorithm, allowing us to successfully learn the bimanual dexterous manipulation policy. To demonstrate the Isaac Gym's efficiency of Bi-DexHands, We provided some results of environmental speed in Table.2 by running on-policy algorithms. Both PPO and HAPPO can achieve more than 20k FPS.

Table 2: Mean and standard deviation of FPS (frame per second) of the environments in Bi-DexHands.

| Algorithms | CatchUnderarm | CatchOver2Underarm | CatchAbreast | TwoCatchUnderarm |
|---|---|---|---|---|
| PPO | $35554 \pm 613$ | $35607 \pm 344$ | $35164 \pm 450$ | $32285 \pm 898$ |
| HAPPO | $23929 \pm 98$ | $23827 \pm 135$ | $23456 \pm 255$ | $23205 \pm 168$ |

## 5.2 RL/MARL results

Currently, we evaluate the performance of PPO, SAC, TRPO, MAPPO, HATRPO and HAPPO algorithms on these 20 tasks, and we implemented the rest of the RL/MARL algorithms in our Github repository. The performance of each algorithm are shown in Figure 3. Note that the experiments of MARL algorithms run based on two agents, which means each hand represents an agent. It can be observed that the PPO algorithm performs well on most tasks. Although there are some tasks that require two-hand cooperation, PPO algorithm is still better than HAPPO, MAPPO algorithms in most cases. This may be because PPO algorithm is able to use all observations for training the policy, while MARL can only use partial observations. However, in most tasks, the more difficult and require the cooperation of both hands, the smaller the performance gap between PPO and HAPPO, MAPPO, indicating that the multi-agent algorithm can improve the performance of bimanual cooperative manipulation. Another finding is that the SAC algorithm does not work on almost all tasks. It may be due to 1) the off-policy algorithm has a lower improvement in high sampling efficiency than on-policy. 2) The policy entropy of SAC brings instability to policy learning under the high-dimension input. We discuss this finding in detail in Appendix C.

## 5.3 Offline RL results

We build offline datasets with four datatypes, *i.e.*, random, replay, medium, and medium-expert. The data collection follows that in D4RL-MuJoCo [68], which is a standard offline benchmark, and the details are given in Appendix A.3. We evaluate behavior cloning (BC), BCQ [59], TD3+BC [58], and IQL [60] on two tasks, Hand Over and Door Open Outward, and report normalized scores in Table 3. BCQ and TD3+BC could obtain significant performance improvement compared with behavior policy (BC). However, the action space and state space in Bi-DexHands are much larger than that in MuJoCo, which means the problem of out-of-distribution action [58] is more severe in Bi-DexHands datasets. That is the reason why IQL could only achieve performance improvement in several datasets. Due to the potential large distribution shift, we believe Bi-DexHands can be a more challenging and meaningful offline benchmark for offline RL research.

## 5.4 Generalization ability

The goals of our generalization evaluation is 1) to find out the ability of current multi-task and meta reinforcement learning algorithms to generalize on the tasks we designed. 2) to find out whether the tasks that are harder for babies are also harder for RL. The previous RL/MARL results have proved that our individual task is solvable. For goal 1), we evaluate the multi-task PPO [47] and ProMP [63] algorithms on MT1, ML1, MT4, ML4, MT20, and ML20. We also provided the results

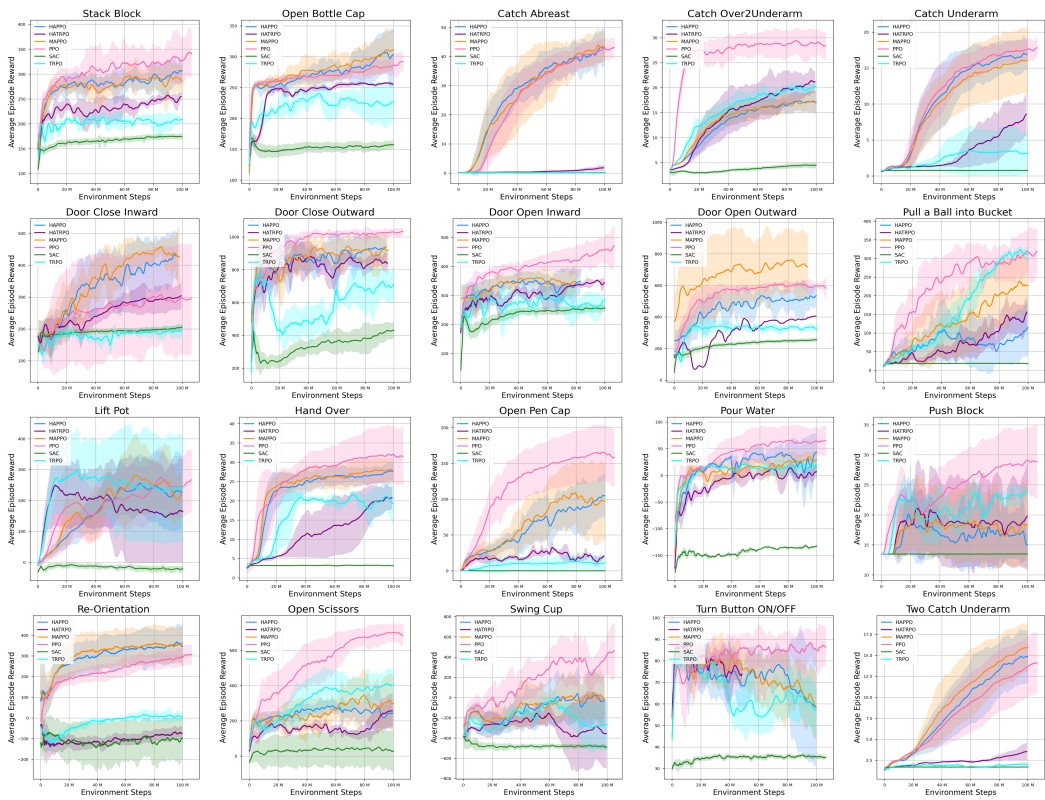

Figure 3: Learning curves for all 20 tasks. The shaded region represents the standard deviation of the score over 10 trials. Curves are smoothed uniformly for visual clarity. All algorithms interact with environments in 100M steps and the number of parallel simulations is 2048.

Table 3: Normalized score in offline tasks.

| Tasks | Datasets | Online PPO | BC | BCQ | TD3+BC | IQL |
|---|---|---|---|---|---|---|
| Hand Over | random | 100.0 | $0.7 \pm 0.2$ | $1.0 \pm 0.1$ | $0.9 \pm 0.2$ | $0.7 \pm 0.4$ |
| | replay | 100.0 | $17.5 \pm 3.5$ | $61.6 \pm 4.9$ | $\mathbf{70.1} \pm 2.1$ | $43.1 \pm 2.3$ |
| | medium | 100.0 | $61.6 \pm 1.0$ | $\mathbf{66.1} \pm 1.9$ | $65.8 \pm 2.2$ | $57.4 \pm 1.5$ |
| | medium-expert | 100.0 | $63.3 \pm 1.4$ | $81.7 \pm 4.9$ | $\mathbf{84.9} \pm 5.3$ | $67.2 \pm 3.6$ |
| Door Open Outward | random | 100.0 | $2.1 \pm 0.6$ | $23.8 \pm 2.9$ | $\mathbf{34.9} \pm 4.3$ | $3.8 \pm 1.0$ |
| | replay | 100.0 | $36.9 \pm 4.3$ | $48.8 \pm 4.5$ | $\mathbf{60.5} \pm 2.6$ | $31.7 \pm 2.0$ |
| | medium | 100.0 | $63.9 \pm 0.7$ | $60.1 \pm 2.3$ | $\mathbf{66.3} \pm 0.7$ | $56.6 \pm 1.2$ |
| | medium-expert | 100.0 | $69.0 \pm 6.4$ | $\mathbf{73.7} \pm 4.5$ | $71.9 \pm 3.5$ | $53.8 \pm 1.8$ |

of random policy and using the PPO algorithm in individual task as the ground truth for comparison. The average reward for each training is shown in Table 4. We can observe that the multi-task PPO does not perform well, and the ProMP have tiny performance improvement compared with random policy. It may because it's hard to learn policy from individually each task itself in Bi-DexHands. Therefore, we still have a lot of room to improve the generalization ability of bimanual dexterous hands under cross-task setting, which is a meaningful open challenge for the community.

Table 4: The average reward of all tasks for MT1, ML1, MT4, ML4, MT20, and ML20 on 10 seeds.

| Method | MT1 | MT4 | MT20 | Method | ML1 | | ML4 | | ML20 | |
|---|---|---|---|---|---|---|---|---|---|---|
| | | | | | train | test | train | test | train | test |
| Ground Truth | 15.2 | 24,3 | 32.5 | Ground Truth | 15.0 | 15.8 | 28.0 | 13.1 | 33.7 | 26.1 |
| Multi-task PPO | 9.4 | 5.4 | 8.9 | ProMP | 0.95 | 1.2 | 2.5 | 0.5 | 0.02 | 0.36 |
| Random | 0.61 | 1.1 | -2.5 | Random | 0.59 | 0.68 | 1.5 | 0.24 | -2.9 | 0.27 |

For goal 2), we use random and ground truth reward to normalize the results of all tasks in MT20 and arrange them in the order of increasing age. The results is shown in Fig.4. It can be seen that

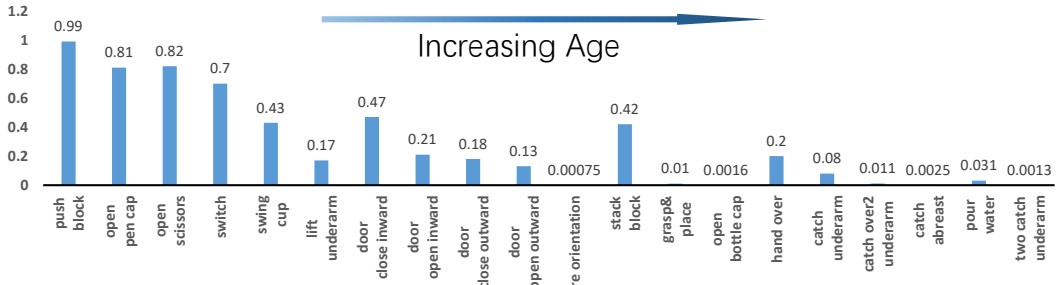

Figure 4: The normalized reward run by the MTPPO algorithm under the MT20 setting. The tasks from left to right according to the increase of corresponding age. The normalized score is computed by score $= \frac{\text{reward-random reward}}{\text{ground truth reward-random reward}}$.

in general, as the age of the person corresponding to the task increases, the difficulty for RL also increases, which proves that our task design is designed with rationality and relevance to people.

## 6 Conclusion and Future Work

We introduced a benchmark, Bi-DexHands, which consists of well-designed tasks and a large variety of objects for learning bimanual dexterous manipulation. We investigated the motor development process of infants' dexterity from cognitive science, and carefully designed more than twenty tasks for RL based on the results, hoping that robots can learn dexterity like humans. With the help of the Isaac Gym simulator, it can run thousands of environments in parallel, improving the sample efficiency for RL algorithms. Moreover, the implemented RL/MARL/offline RL algorithms achieve superior performance on tasks with simple manipulation skills required. Meanwhile, complex manipulations still remain challenging. In particular, when the agent is trained to master multiple manipulation skills, the results of multi-task/Meta RL are not satisfactory. Interestingly, we found that under the multi-task setting, RL exhibited results associated with the development of human intelligence, that is, the trend of RL performance matches with the development of human ages. So far, in bimanual dexterous robot hand manipulation, the current reinforcement learning can reach the level of 48-months infants.

However, we think that the limitation of Bi-DexHands is that it does not support the deformable object manipulation tasks. Dexterous hands have unique advantages in manipulating deformable objects, but our tasks currently only cover articulated rigid body object manipulation. We hope to develop in this direction in the future. Another limitation is that our tasks primarily train with policies with a state-based observation space, which is difficult for sim-to-real transfer because such inputs are not available in the real world. Our work will advance the field of robotics, increasing the level of automation in factories or lives to replace human labor. The development of this field will reduce the need to put humans in dangerous situations and improve the quality of human life, but it will also bring about the potential for worker displacement.

We identify four main future directions toward mastering human-level bimanual dexterous manipulation. **1)** Learning from demonstration: our platform needs some human teaching data to study learning from demonstration. **2)** Soft body and deformable objects simulation: we need a better physics engine to support our research on software and task design, to be more specific daily life scenes. **3)** Current meta/multi-task RL algorithms are unable to perform all tasks in our benchmark successfully, which calls for substantial further development on the algorithmic design end. **4)** We would like address the sim-to-real gap by transferring the simulation result on real dexterous hands. In particular, we hope our benchmark results can serve as a start point to help researchers transfer RL-learned skills to reality and help real-world robots to learn dexterous manipulation.

## 7 Acknowledge

This work is partly sponsored by Beijing Municipal Science & Technology Commission (Project ID: Z221100003422004)

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
