# OpenReview forum: "Towards Human-Level Bimanual Dexterous Manipulation with Reinforcement Learning"
_NeurIPS.cc/2022/Track/Datasets_and_Benchmarks — NeurIPS 2022 Datasets and Benchmarks _

### Official Review · Reviewer_PbRo · 2022-07-23
**A limited but welcome contribution to robotic benchmarks of current interest.**

**Rating:** 7
**Confidence:** 4
**Correctness:** The datasets and evaluations look rea…

**Strengths:**

The benchmark is highly relevant to current work in applying the various RL approaches to robotics challenges.

The simulator is very fast, particularly since observations are state-based only. Image-based observations will decrease performance.

The reported experiments are sufficient to show the benchmark’s usefulness for continued research. The most simple tasks can be solved with standard RL techniques, given state observations, along with enough computation and reward shaping.

The paper presents a good analysis of relative PPO vs. SAC performance.


**Weaknesses:**

The benchmark currently supports only state-based observations, which greatly limits its utility to the community. But this is a reasonable first step to take, and it should be possible to add image-based observations in the future.

Many of the RL algorithms provided have not yet been tested, so bugs remain.

Support for applying the algorithms to physical shadow hands would be a significant extension of the framework.

Generally, the benchmark is still a work in progress. This is not too surprising, given its ambitious breadth. Still, Bi-DexHands is a solid initial step, and will hopefully become more complete soon.


**Additional Feedback:**

It would be helpful to include pointers and comments related to the physical shadow hands, and how closely they may or may not relate to the shadow hands simulated here.

**Clarity:**

The documentation and the paper are fairly clear and easy to follow. Community involvement can be expected to drive improvements.

**Documentation:**

I failed to install Isaac Gym to test the benchmark. But the repository looks good, and is already receiving feedback from the community.

**Ethics:**

No concerns.

**Relation To Prior Work:**

The relevant prior work is covered well in the paper.

**Summary And Contributions:**

The Bi-DexHands Benchmark is a new Isaac Gym simulator that supports experiments with bi-dexterous robotic manipulation. It covers a wide range of settings (Single-agent/Multi-agent RL, Offline RL, Multi-task RL, and Meta RL), and includes a variety of tasks requiring two hands, along with several implementations of corresponding algorithms. Baseline results exercise the simulator, and demonstrate significant research challenges left in bi-dexterous manipulation.

---

> ### Author Response · Authors · 2022-08-13
> **Response to reviewer PbRo**
>
> We would like to sincerely thank you for reviewing our paper and providing constructive comments! We appreciate your enthusiasm for the work!
>
> Here we address your comments below:
>
> > The benchmark currently supports only state-based observations, which greatly limits its utility to the community. But this is a reasonable first step to take, and it should be possible to add image-based observations in the future.
>
> Yes, I agree with you very much. Regarding visual input, we have also tried it. Isaacgym can use RGBD cameras to provide us with visual information, which can be directly used as image input or processed into other modal data such as point clouds. But there is a problem is that the parallelism of Isaac Gym's cameras is not very good. It can only obtain images one by one env serially, which will greatly slow down the running speed. At the same time, the training of the dexterous hand is very difficult and greatly depends on the high sampling efficiency, so we do not use the input of other modalities. But this is undoubtedly very important, so we provide an example for other modal (point cloud) inputs for future exploration or development. See [here](https://github.com/PKU-MARL/DexterousHands/blob/main/bi-dexhands/tasks/shadow_hand_point_cloud.py).
>
> > Many of the RL algorithms provided have not yet been tested, so bugs remain.
>
> Yes, you are right. Because we are a comprehensive RL benchmark, including RL, MARL, MTRL, MLRL, OffRL and so many tasks, it is very difficult to benchmark all algorithms. But we are working hard in this direction. At present, the TRPO, HATRPO algorithm has been newly evaluated, check our revision paper. But since TRPO-type algorithms need to solve the hessian matrix, which takes a long time, So we only completed the benchmark for part of the task. We will try to run through the rest of the time and continue to update during the rebuttal period.
>
> > Support for applying the algorithms to physical shadow hands would be a significant extension of the framework.
>
> Yes, sim2real is a very important topic. As we mentioned in the future work in the paper, we are also working in this direction and hope to one day transfer our trained models to the real world.
>
> > Generally, the benchmark is still a work in progress. This is not too surprising, given its ambitious breadth. Still, Bi-DexHands is a solid initial step, and will hopefully become more complete soon.
>
> Thank you for your understanding! Yes, Bi-DexHands is still in a relatively early stage, and we hope that we can participate with the community and continue to develop it. We have written a Todo list, see [here](https://github.com/PKU-MARL/DexterousHands#future-plan).
>
> > It would be helpful to include pointers and comments related to the physical shadow hands, and how closely they may or may not relate to the shadow hands simulated here.
>
> Yes, the connection between the Shadow Hand in simulation and reality is very important. First of all, the physics simulation effect of PhysX used by Isaac Gym is very good, especially in contact rich. That's one of the reasons why we chose Isaac Gym as our platform. Second, our Shadow Hand model is the same as the Shadow Hand model in OpenAI Gym, which is very similar to the physical Shadow Hand. We also support custom physical parameters such as stiffness, damping, effort, etc., which helps us to study sim2real problems. Finally, the powerful parallelization capability of Isaac Gym can perform efficient domain randomization, which is very important for sim2real transfer.

---

> > ### Comment · Reviewer_PbRo · 2022-08-20
> > **Reply**
> >
> > Thank you for the helpful responses and continuing to expand the framework!

---

> > > ### Author Response · Authors · 2022-08-26
> > > **Thank you for your appreciation**
> > >
> > > Thank you very much for your appreciation. We will ensure long-term maintenance of Bi-DexHands and put in more effort to further improve it.

---

### Official Review · Reviewer_wDmx · 2022-07-26

**Rating:** 7
**Confidence:** 3

**Strengths:**

1. This paper introduces a comprehensive benchmark with 13 bimanual dexterous manipulation tasks of varying complexity, based on different levels of human motor skill development. The authors benchmark different baselines covering reinforcement learning, multi-agent RL, offline RL, multi-task RL, and meta-RL. The code and tasks are well-documented, such as details on observations, actions, and rewards for all tasks in the Appendix.

2. The suite of tasks is built off Isaac Gym, enabling efficient scaling by using GPU-based physics simulation. This lowers the barrier for those with smaller compute-budgets to run experiments.


**Weaknesses:**

1. The paper is missing an upfront discussion of limitations, as well as a broader impact statement. These do not appear in the main body, or the Appendix. While some limitations are mentioned in Section 6, it is in the context of future directions.
2. The paper would benefit from improving the clarity of the writing, see Additional Feedback section.


**Additional Feedback:**

L66: Can you clarify what “enormous target objects” means?

L267: Does “MuJoCo” mean the “D4RL-MuJoCo” tasks?

L267-269: Can you what is meant by “out-of-distribution action is more severe”? What is the distribution shift?

For Table 4, is it difficult to gauge the success of methods by averaging reward across tasks. Is there some measure of progress or success for each of the tasks? Even a normalized measure would be better, similar to Figure 4.

L206-207: I appreciate the analogies of task difficulty to human measures. However, I think analyzing the performance of different RL algorithms on the proposed benchmark is more nuanced than “precisely point out agents’ intelligence level”.

---

Minor comments (this is not a comprehensive list, and a thorough editing pass is recommended):

L9: “Issac Gym”, typo

L24: “the human hands”, grammar

L45: “og”, typo

L109: “toward master”, grammar

L253-255: grammar

The paper occasionally uses “Shadow hand”, which should probably be capitalized to “Shadow Hand”?

L210: Should be capitalized as “Meta-World”


**Clarity:**

The paper can be difficult to understand at times, and would benefit from a thorough editing readthrough. See Additional Feedback.


**Correctness:**

To my knowledge, yes.


**Documentation:**

Yes, all the tasks are thoroughly described. The appendix contains detailed tables covering the action and observation space of all the tasks. The appendix also describes baselines methods and includes accompanying hyperparameters.


**Ethics:**

N/a.


**Relation To Prior Work:**

Yes, to my knowledge, no prior work has presented a simulated suite of tasks for bimanual dexterous manipulation.

The related work section would benefit from additional discussion of prior work on single-hand dexterous manipulation [1, 2, 3, 4] and cooperative multi-agent environments, such as [5].

1. https://arxiv.org/abs/2111.03062
2. https://arxiv.org/abs/2108.05877
3. https://arxiv.org/abs/2203.06173
4. https://arxiv.org/abs/2204.12490
5. https://arxiv.org/abs/2105.12196


**Summary And Contributions:**

This paper introduces Bi-DexHands, a simulation suite for bimanual dexterous hand manipulation that is built in Isaac Gym, enabling fast GPU-accelerated physics simulation. The authors benchmark a comprehensive set of different approaches covering reinforcement learning, multi-agent RL, offline RL, multi-task RL, and meta-RL. Their baseline results show that while PPO-based algorithms can perform well at single-task settings, multi-task learning still poses a considerable challenge for current algorithms on Bi-DexHands.

---

> ### Author Response · Authors · 2022-08-13
> **Response to reviewer wDmx**
>
> We would like to sincerely thank you for reviewing our paper and providing constructive comments!
>
> Here we address your comments below:
>
> > The paper is missing an upfront discussion of limitations, as well as a broader impact statement. These do not appear in the main body, or the Appendix. While some limitations are mentioned in Section 6, it is in the context of future directions.
>
> Thank you for reminding. We have added a discussion of limitations in section 6, see our revision paper.
>
> > The paper would benefit from improving the clarity of the writing, see Additional Feedback section.
>
> We have revised and replied to the writing and some unclear expressions in the revision according to the suggestions in additional feedback. Thank you for your help in improving our paper!
>
> > Yes, to my knowledge, no prior work has presented a simulated suite of tasks for bimanual dexterous manipulation.
> The related work section would benefit from additional discussion of prior work on single-hand dexterous manipulation [1, 2, 3, 4] and cooperative multi-agent environments, such as [5].
> https://arxiv.org/abs/2111.03062
> https://arxiv.org/abs/2108.05877
> https://arxiv.org/abs/2203.06173
> https://arxiv.org/abs/2204.12490
> https://arxiv.org/abs/2105.12196
>
> We have discussed the above 5 jobs in the relative work. At the same time, we are also investigating more work related to dexterous manipulation, so your suggestion is very useful, thank you!
>
> > L66: Can you clarify what “enormous target objects” means?
>
> To put it simply, the Bi-DexHands object-catching task uses objects from the YCB dataset, while other tasks use objects from the Sapien dataset. Bi-DexHands can be customized to use objects from the YCB and Sapien dataset in place of existing items, so we say we have enormous target objects. For details, see our newly written help document [here](https://github.com/PKU-MARL/DexterousHands/blob/main/docs/customize-the-environment.md#How_to_replace_objects_in_existing_tasks_with_items_from_YCB_and_Sapien_datasets).
>
> > L267: Does “MuJoCo” mean the “D4RL-MuJoCo” tasks?
>
> Yes, it refers to D4RL-MuJoCo under the setting of Offline RL, which we have corrected in the paper.
>
> > L267-269: Can you what is meant by “out-of-distribution action is more severe”? What is the distribution shift?
>
> Distribution shift generally refers to the difference between the training distribution and the test distribution in supervised learning, and refers to the inconsistency between the training policy and the behavior policy in offline reinforcement learning (i.e. our policy might be trained under one distribution, but it will be evaluated on a different distribution). Compared with other robot-control RL environments, our Bi-DexHands have higher action dimensions, so it is more probably encounter data that is not in training distribution during testing, so it is more seriously affected by distribution shift.
>
> > For Table 4, is it difficult to gauge the success of methods by averaging reward across tasks. Is there some measure of progress or success for each of the tasks? Even a normalized measure would be better, similar to Figure 4.
>
> Yes, using average reward is not very intuitive. But there is a problem that some of our tasks are still difficult to complete, providing to the community as a challenge. So using the success rate directly may not be a good measure of these difficult tasks. Therefore, we are considering using the reward after normalization as a performance indicator. It has been added to our Todo list, see [here](https://github.com/PKU-MARL/DexterousHands#future-plan).
>
> > L206-207: I appreciate the analogies of task difficulty to human measures. However, I think analyzing the performance of different RL algorithms on the proposed benchmark is more nuanced than “precisely point out agents’ intelligence level”.
>
> Yes, analyzing the performance of different RL algorithms on the proposed benchmark makes a lot of sense. We present an analysis of the effects of the SAC and PPO algorithms in the Appendix C, although there may still be room for discussion. Because we are a comprehensive RL benchmark, including RL, MARL, MTRL, MLRL, OffRL and so many tasks, it is very difficult to benchmark all algorithms. But we are working hard in this direction. At present, the TRPO, HATRPO algorithm has been newly evaluated, check our revision paper. But since TRPO-type algorithms need to solve the hessian matrix, which takes a long time, So we only completed the benchmark for part of the task. We will try to run through the rest of the time and continue to update during the rebuttal period.
> > Minor comments
>
> All corrected, thanks for your help!

---

### Official Review · Reviewer_TPhv · 2022-07-26
**Solid benchmark for bimanual dexterous manipulation; room for improvement for related work, task setup, and experimental evaluation.**

**Rating:** 7
**Confidence:** 4

**Strengths:**

The main contribution is threefold: a) relatively comprehensive (bi-manual) dexterous hand manipulation benchmarks, b) very fast simulation speed, c) comprehensive baselines methods across single/multi-agent RL, offline RL, and multi-task/meta RL. It’s pretty relevant for the community that is interested in (bi-manual) dexterous hand manipulation. The benchmark is open-souced.


**Weaknesses:**

The paper does have a few major weaknesses. It motivates the benchmark by arguing that the tasks are designed to match different levels of human motor skills according to cognitive science literature. However, this claim seems to be built upon some shaky ground. The translation from “human skill” to “robotic task” seems a bit arbitrary. “Child places 10 pellets in the bottle” got translated into “put a ball into a bucket”. “Infant differentiate individual finger movement” got translated into “reorient a Rubrik's cube”. It’s very much unclear how these tasks are constructed and whether they really capture the motor skills of infants and children. Secondly, Two Catch Underarm was labeled as “adult” while it’s reasonable to assume many dexterous children/teenagers could do it. Also, in the “conclusion” section, the authors claimed that “the current reinforcement learning can reach the level of 48-months infants”, which is a very dramatic over-generalization. A 48-month infant can do way more tasks than pouring water and catching objects. Finally, a less important point, quite a few tasks are not inherently bi-manual, so it seems very forced to have both hands there (e.g. re-orientation, stack block, even door opening, etc)

**Additional Feedback:**

The supplementary video is helpful. Thank you.

**Clarity:**

The paper is relatively well-written, but the overall text contains a large number of typos and grammatical errors. The task names are also not consistent throughout the paper (compare table 1, table 2, figure 3, and figure 5 in the appendix and Appendix A.2 Detailed components of tasks).


**Correctness:**

There are a few questions regarding the correctness of the dataset. Firstly, it’s unclear how the authors propose to leverage the “thousands of objects” from SAPIEN dataset. In the paper, only ten to twenty objects were observed in the task. It’s still unclear how someone can use different object models for these tasks (e.g. for pouring water, how to use different models for the pot and the cup?). On a related note, the task setup seems to be specific for each object model. For example, in the “Lift Pot” task, the reward function relies on the “distance from the left hand to the left handle”. However, different models of the pot category can have a different number of handles and the handles can be located at different positions. Are these locations programmatically accessible from SAPIEN? It’s unlikely the case. Also, many tasks require a notion of the “goal pose”, which can also depend on the exact object models used. For example, in the “Two Catch Underarm” task, the goal poses probably should depend on the size of the objects being thrown. In general, it’s very unclear how these task setups can be object-agnostic, i.e. no need to hard-code these “handle positions” and “goal poses” manually.

Furthermore, the authors implemented 5 single-agent RL algo., 5 multi-agent RL algo., 3 multi-task RL algo. and 2 meta-learning RL algo., which is really great and comprehensive. However, in experimental evaluation, only 2 single-agent RL algo., 2 multi-agent RL algo., 1 multi-task RL algo., and 1 meta-learning RL algo. are evaluated. It would be great if the authors can benchmark the performance of all the baselines that they implemented and include the additional results in the rebuttal.

The current experimental results that the authors presented also raise a few questions. First of all, it’s not convincing why the SAC does not work on almost any tasks. The authors tried to offer their hypothesis in Section C of the appendix, but it’s still not 100% convincing. For example, from Figure 6 of the appendix, it seems that the SAC training becomes less stable when the number of parallel environments increases, which is very counter-intuitive. The authors should consider significantly increasing the replay buffer size of SAC (currently it’s only 5000, as specified in Table 43 in the appendix). Since the observation space only contains states (not images, more compact), so this should be doable. Also, maybe increasing batch size would help. Finally, the meta-RL baseline ProMP completely fails to solve the tasks, even in the simple scenario of ML1, in which the tasks clearly share some underlying structure. It’s recommended that the authors double-check the implementation of ProMP for any potential issues.


**Documentation:**

The authors listed all the relevant URLs in the paper. Everything seems to be open-sourced under Apache License 2.0.


**Ethics:**

In the checklist, the authors replied Yes to the question “Did you discuss any potential negative societal impacts of your work”, but none can be found. Maybe somehow I missed it?


**Relation To Prior Work:**

In the related work section, the authors mentioned previous learning-based and non-learning-based methods that tackle dextrous hand manipulation and also robot manipulation in general. However, they didn’t compare their benchmark with other robotic manipulation benchmarks (e.g. robosuite, RLBench, metaworld, etc). These benchmarks are slightly different because they don’t use dexterous hands, but it’s still relevant enough to be included and compared.



**Summary And Contributions:**

The paper proposes the Bimanual Dexterous Hands Benchmark (Bi-DecHands) which involves two dexterous hands and twenty manipulation tasks. The authors demonstrate efficient RL training, thanks to Issac Gym, and benchmark SOTA single-agent/multi-agent RL, offline RL, multi-task RL, and meta RL on these tasks.

---

> ### Author Response · Authors · 2022-08-13
> **Response to reviewer TPhv (1/3)**
>
> We would like to sincerely thank you for reviewing our paper and providing constructive comments!
> Here we address your comments below:
>
> > The paper does have a few major weaknesses. It motivates the benchmark by arguing that the tasks are designed to match different levels of human motor skills according to cognitive science literature. ... It’s very much unclear how these tasks are constructed and whether they really capture the motor skills of infants and children.
>
> Thank you for your interest in the detailed design of our tasks! Yes, translation from “human skill” to “robotic task” is very difficult. We try to design the task according to the behavior of the baby at this age based on being able to complete it. For example, we learned that Child places 10 pellets in the bottle at the age of 22-28 month. On this basis, we abstracted the action of pick and place, that is, we designed the task of putting a ball into a bucket. The main requirement of reorienting a Rubrik's cube is the dexterity of each finger, which can well reflect the ability of Infant to differentiate individual finger movement, so we chose this task as a representative of this age stage.
>
> > Secondly, Two Catch Underarm was labeled as “adult” while it’s reasonable to assume many dexterous children/teenagers could do it.
>
> The Two Catch Underarm task is somewhat similar to the juggling ball, and the average adult or teen still needs to try or train to complete it well. Therefore, we label this behavior that obviously require more advanced intelligence as an adult, although it is true that some teenagers can also complete this task.
>
> > Also, in the “conclusion” section, the authors claimed that “the current reinforcement learning can reach the level of 48-months infants”, which is a very dramatic over-generalization. A 48-month infant can do way more tasks than pouring water and catching objects.
>
> Indeed, a 48-month infant can do way more tasks than pouring water and catching objects. But according to the theory of developmental behavior, these two behaviors belong to a 48month baby's behavior. In other words, being able to complete these tasks marks a 48-month baby's level of manipulation, and that's why we say that.
>
> > Finally, a less important point, quite a few tasks are not inherently bi-manual, so it seems very forced to have both hands there (e.g. re-orientation, stack block, even door opening, etc)
>
> Yes, we also encountered this problem in the design task. Based on developmental-behavioral research, we try our best to think and design tasks that require the cooperation of both hands. At the same time, there are inevitably some tasks that can be done with one hand. But we think there are times when two hands are better at helping you do certain tasks that you can do with one hand, or even learn a completely different policy. For example, with stack blocks, using both hands can learn policy that can not be done with one hand: when one hand tries to stack a block on top of another block, the other hand can adjust the block below to fit the position of the block above, better complete the task. On the other hand, some actions such as opening the door are indeed done by each hand, but we still think that they are meaningful. Because these robotic tasks need to be used to correspond to the tasks that the baby of the age can complete, and the hands are more human-like settings after all.

---

> > ### Author Response · Authors · 2022-08-13
> > **Response to reviewer TPhv (2/3)**
> >
> > > There are a few questions regarding the correctness of the dataset. Firstly, it’s unclear how the authors propose to leverage the “thousands of objects” from SAPIEN dataset. ... In general, it’s very unclear how these task setups can be object-agnostic, i.e. no need to hard-code these “handle positions” and “goal poses” manually.
> >
> > Thanks for your pointing out! Yes, customizable tasks are very important to the development of a platform. We used objects from the YCB dataset for the object-catching task, while objects from the Sapien dataset were used for other tasks. Bi-DexHands can not only customize the use of objects in the YCB and Sapien datasets to replace existing items, but also support custom tasks more conveniently. The YCB+Sapien dataset that can be used can probably have a total of "thousands of objects" choices. To facilitate this, we have written an additional markdown file to help users define their own tasks using Bi-DexHands, see [here](https://github.com/PKU-MARL/DexterousHands/blob/main/docs/customize-the-environment.md#How_to_create_your_own_task).
> > Regarding how to use different objects, except that the object-catching task is object-agnostic, other tasks do need to be defined according to different objects. But there is not much modification required. For example, for the "Lift Pot" task you mentioned, after replacing the pot in the Sapien dataset, it is necessary to redefine the positions of the left handle and the right handle. This usually requires us to manually set 3-dimensional position information, which is not programmatically accessible from SAPIEN. I made a specific description in [here](https://github.com/PKU-MARL/DexterousHands/blob/main/docs/customize-the-environment.md#How_to_replace_objects_in_existing_tasks_with_items_from_YCB_and_Sapien_datasets).
> > Regarding the goal pose, this is the position and rotation of the center point of an object. If the size is different, the impact is usually not great, as long as the size does not exceed the range that can be in-hand manipulation.
> >
> > > Furthermore, the authors implemented 5 single-agent RL algo., 5 multi-agent RL algo., 3 multi-task RL algo. and 2 meta-learning RL algo., which is really great and comprehensive. ... It would be great if the authors can benchmark the performance of all the baselines that they implemented and include the additional results in the rebuttal.
> >
> > Yes, you are right. Because we are a comprehensive RL benchmark, including RL, MARL, MTRL, MLRL, OffRL and so many tasks, it is very difficult to benchmark all algorithms. But we are working hard in this direction. At present, the TRPO, HATRPO algorithm has been newly evaluated, check our revision paper. But since TRPO-type algorithms need to solve the hessian matrix, which takes a long time, So we only completed the benchmark for part of the task. We will run through the rest of the time and continue to update during the rebuttal period.
> >
> > > The current experimental results that the authors presented also raise a few questions. First of all, it’s not convincing why the SAC does not work on almost any tasks. ... Since the observation space only contains states (not images, more compact), so this should be doable. Also, maybe increasing batch size would help.
> >
> > Regarding why SAC cannot work well, we have been paying attention to this problem for a long time and have double-checked it many times. The result is exactly as stated in the paper, so we are very hopeful to explore the reason. In our setup, the replay buffer size is not strictly 5000, but 5000 x num_env, which will expand as the parallel environment increases. Therefore, when we use 2048 parallel environments in SAC, the actual replay buffer size should be 5000 x 2048, and we need to use all the 24G GPU memory of the RTX3090. In fact, we have also tried reducing the number of parallel environments to increase the replay size, but it still doesn't work well. However, when we adjust the replay buffer on the humanoid task, the effect is not very large. Therefore, we think the reason is not here, our replay buffer size is sufficient. In general, we also consider this phenomenon counter-intuitive, but the experimental results do. We are confident that our SAC has been checked many times without a problem, so we are very hopeful that the community will explore this problem with us.

---

> > > ### Author Response · Authors · 2022-08-13
> > > **Response to reviewer TPhv (3/3)**
> > >
> > > > Finally, the meta-RL baseline ProMP completely fails to solve the tasks, even in the simple scenario of ML1, in which the tasks clearly share some underlying structure. It’s recommended that the authors double-check the implementation of ProMP for any potential issues.
> > >
> > > Thank you for reminding! Indeed, the reward obtained by ProMP is not so good, but in fact, it can probably complete the task. Our ML1 uses the reward design of the exponential function, and a little distance will have a great impact on the reward when approaching the target point. When we render the training, we can see that in general, the object can be thrown to the other hand, but the perfect coincidence with the target may be less, and the reward feels much worse under the exponential function. But your consideration is also meaningful because we need to adapt to the parallelism simulation of Isaac Gym, we have completely re-implemented the meta RL algorithm, and there may indeed be potential issues. We have re-double-checked to ensure that there is no problem in the implementation.
> > >
> > > > The paper is relatively well-written, but the overall text contains a large number of typos and grammatical errors. The task names are also not consistent throughout the paper (compare table 1, table 2, figure 3, and figure 5 in the appendix and Appendix A.2 Detailed components of tasks).
> > >
> > > Thanks for your pointing out. We have checked and synced all task names, but there may be some misunderstandings. To be precise, we have 20 different tasks. The 13 tasks in table 1 are the stages of infant motor development that we referenced when designing the tasks. There are some tasks of the same type that belong to the same stage, so we merge them together. For example, opening and closing the door, we only use one column to introduce, but there are actually 4 different tasks: **DoorOpenInward**, **DoorCloseInward**, **DoorCloseOutward**, **DoorOpenOutward**; we only use one column for object-catching task, and there are actually 5 tasks: **Hand Over**, **Catch Underarm**, **Catch Over2Underarm** **Catch Abreast**, **TwoCatch Underarm**, which appear in table 2, figure 3, and figure 5.
> > >
> > > > In the related work section, the authors mentioned previous learning-based and non-learning-based methods that tackle dextrous hand manipulation and also robot manipulation in general. .... These benchmarks are slightly different because they don’t use dexterous hands, but it’s still relevant enough to be included and compared.
> > >
> > > Thanks for your comment. We have discussed and compared these benchmarks in relative work, see our revision paper.
> > >
> > > > In the checklist, the authors replied Yes to the question “Did you discuss any potential negative societal impacts of your work”, but none can be found. Maybe somehow I missed it?
> > >
> > > Thanks for your comment! We are sorry for accidentally leaving out this question at the time. We've added the discuss societal impacts in the Section. 6, check our revision paper. We are very sorry for the inconvenience caused to you.

---

> ### Comment · Reviewer_TPhv · 2022-08-16
> **Discussion with Authors**
>
> Dear Authors,
>
> Thank you for your detailed response to my review and for leaving ample time for discussion.
>
> > Customize the environment
>
> Thank you for providing the additional markdown file about how to customize the environments (e.g. objects) for a given task or to create my own task. This is very helpful and illustrative. From my understanding, if the user needs to use a new object from SAPIEN, they might need to adjust 1) the size/scale of the object, 2) initialization positions of the object, 3) auxiliary grasping point, 4) specifying the correct dofs, etc. From the perspective of a user who simply wants to randomize objects for a task for generalization purposes, it's probably too much work and too error-prone. Although it's impossible to do within the short window of rebuttal, I would encourage the authors to provide these annotations / meta information for the objects in YCB+SAPIEN dataset, as part of the benchmark, maybe in the next version. The user should be able to change the object by simply specifying the object model ID in the config file. It's going to be quite some work, but I believe it will maximize the impact of the proposed benchmark.
>
> > Additional evaluation result for all the baselines implemented on all tasks.
>
> Sounds good. I will watch out for more results coming up.
>
> > SAC implementation
>
> Okay I am convinced that 5000 x 2048 should be sufficient for the replay buffer size. I guess the problem lies somewhere else and hopefully the community can figure out why when they use your benchmark.
>
> > ProMP implementation
>
> Sorry, I am not sure I understand your comments. Have you double-checked your implementation and found no issues?
>
> > Syncing task names across the paper
>
> Thanks for explaining the details about the naming. Please consider changing "Task Name" to "Human Task Name" in Table 1 for better clarity. Also, in Fig 3, the title of each subplot should exactly match the task name.
>
> Thanks!

---

> > ### Author Response · Authors · 2022-08-19
> > **Discussion with Reviewer TPhv**
> >
> > Thank you for your comment! we are glad to discuss with you to make our work better.
> >
> > > Thank you for providing the additional markdown file about how to customize the environments (e.g. objects) for a given task or to create my own task. ... It's going to be quite some work, but I believe it will maximize the impact of the proposed benchmark.
> >
> > Thank you for your comment on maximizing the impact of our benchmark! Yes, the main custom step is those four points you said. We also think that this will bring a lot of work and debugging to the user to customize the environment. We will work on improving the convenience of this part in the future.
> >
> > As you comment, since each object in the Sapien dataset has its unique parameters in the environment, manually providing annotations/meta information for each object is a good solution. But indeed it may not be done during rebuttal. So I added this part of the work to our Todo list and continued to work on it. We believe it will be a very important feature when completed and will make our work more solid!
> >
> > > Additional evaluation result for all the baselines implemented on all tasks.
> >
> > Thank you for your concern! We are currently supplementing the baselines, and we will notify you and all reviewers when the results come out.
> >
> > > Okay I am convinced that 5000 x 2048 should be sufficient for the replay buffer size. I guess the problem lies somewhere else and hopefully the community can figure out why when they use your benchmark.
> >
> > Yes, SAC anomalies have always been our concern, but we still think our implementation is fine. In fact, there are already other Isaac Gym projects in our team using our SAC code even better than our PPO. The only difference is that they use RL to control the robotic arm, and the number of environments and action spaces are relatively small. This is consistent with the experimental performance of Humanoid we have done in the appendix, which further confirms our conjecture. Therefore, we believe that our SAC implementation is fine, and we also hope that the community can participate in exploring this phenomenon with us. I believe it will be a very meaningful discovery (RL under high sample efficiency).
> >
> > > Sorry, I am not sure I understand your comments. Have you double-checked your implementation and found no issues?
> >
> > We are very sorry for the inconvenience caused to you. Yes, we have double-checked that our ProMP algorithm is correct. Our ProMP is re-implemented using the [TorchOpt](https://github.com/metaopt/TorchOpt) library to match isaacgym's parallelism. TorchOpt is a high-performance optimizer library built upon PyTorch for easy implementation of functional optimization and gradient-based meta-learning, and provides powerful algorithmic reimplemented. Since the benchmark track is a single-blind review, it allows us to show that the author of TorchOpt is one of the authors of our paper, helped us re-implement ProMP, and performed multiple double-checks. So we think our ProMP is OK.
> >
> > > Thanks for your suggestion! we will change "Task Name" to "Human Task Name" in Table 1 for better clarity and correct the task name in Fig. 3, see our revision paper.
> >
> > Thanks for your suggestion! we will change "Task Name" to "Human Task Name" in Table 1 for better clarity and correct the task name in Fig. 3, see our revision paper.

---

> > > ### Comment · Reviewer_TPhv · 2022-08-25
> > > **Thank you**
> > >
> > > Thank you for your detailed response and the additional experiments during the rebuttal phase. I do not have any further questions and I have bumped up the rating.

---

> > > > ### Author Response · Authors · 2022-08-26
> > > > **Thank you for your appreciation**
> > > >
> > > > Thank you very much for your appreciation. We will ensure long-term maintenance of Bi-DexHands and put in more effort to further improve it.

---

> > ### Author Response · Authors · 2022-08-19
> > **Discussion with Reviewer TPhv: more results have been completed**
> >
> > We are glad to inform you that our additional evaluation result for all the baselines implemented on all tasks has been completed. We newly added TRPO, and HATRPO baseline on 20 tasks, see our revision paper (Fig. 3).

---

### Official Review · Reviewer_5Yjg · 2022-07-27
**Review of paper 'Towards Human-Level Bimanual Dexterous Manipulation with Reinforcement Learning'**

**Rating:** 6
**Confidence:** 4

**Strengths:**

As claimed in the paper, I agree that the proposed work has the following strengths:
1. Support most kinds of RL algorithms in one platform (this will definitely broaden the impact and benefit multiple RL directions while also provide a testbed to benchmark different methods)
2. Highly paralleled, able to sample actions and train efficiently on GPU (they describe in the appendix the improvement of running in parallel on one GPU instead of multiple ones required by Issac Gym originally)
3. Provide a set of objects to test task generalizability over target objects (I believe this generalization is essential to extend RL algorithms to a broader use case that it could possibly adapt to unseen objects after being trained on a publlic object set)



**Weaknesses:**

1. Sensor simulation: I don't understand why this platform only provides point cloud given by depth camera. Other modalities such as RGB, normal, shading, etc. from multiple, configurable views should also be provided, to explore direct end-to-end training from raw sensor input, rather than known object target information like their 6D poses. This point should be an easy add-on though.
2. Ease of task customization: I read through the paper and appendix but didn't find detailed descriptions on designing customized tasks. The authors only mention that objects from provided YCB and SAPIEN datasets could be used to replace the current basis, but how to generate a ground truth target pose or trajectory for every new object, or even a completely new task, is not given. This point is essential for users who want to develop based on this platform and create their own specific tasks.
3. Different reward design for RL: In the appendix, each of 20 tasks is detailed with a given reward function with some hardcoded numbers. I wonder how are these numbers obtained by the authors and whether there are pre-experiments testing different types of rewards, because the reward function design is important to RL and the design details should be explored.
4. The evaluation of some RL algorithms are not finished yet.

**Additional Feedback:**

This work is really a good contribution to the community. Besides the points I mentioned in weakness section, I am curious on several points on further extending this work:
1. How good is the physical contact rich actions simulated in the proposed platform, or Issac Gym?
2. How hard is it to import other dexterous manipulators and design new tasks to the workspace?
3. How to further evaluate the trained RL agents in real world?

**Clarity:**

The paper is well written overall. I recommend a proof-reading on grammars and put more essential details about the dataset itself to the main paper from supplementary materials (a summary of A.1 and A.2)

**Correctness:**

The submission is constructed in a sound way. I got most questionable points answered in the supplementary material.

**Documentation:**

The documentation on github is sufficient.

**Ethics:**

There is no ethics concern of this submission.

**Relation To Prior Work:**

Yes, this work is novel and different from previous works. I recommend the authors considering more diverse tasks and algorithms mentioned in these previous works:

https://ieeexplore.ieee.org/stamp/stamp.jsp?arnumber=8989777
https://ieeexplore.ieee.org/stamp/stamp.jsp?arnumber=9196527
https://ieeexplore.ieee.org/stamp/stamp.jsp?arnumber=9035030
https://arxiv.org/pdf/1904.11382.pdf
https://arxiv.org/pdf/2010.08587.pdf

**Summary And Contributions:**

This paper contributes a novel bi-manual dexterous manipulation benchmark for RL. The highlight is the elaborated design of 20 tasks that covers different difficulty levels equivalent to growth month of a baby, as well as the highly paralleled running platform for RL training based on the new simulator Issac Gym. The authors also provide RL baseline implementations and evaluate them based on categories. It's great the authors make the work open-source already and has got 20 forks and 177 stars on github, showing a wide impact.

---

> ### Author Response · Authors · 2022-08-13
> **Response to reviewer 5Yjg (1/2)**
>
> We would like to sincerely thank you for reviewing our paper and providing constructive comments!
>
> Here we address your comments below:
>
> > Sensor simulation: I don't understand why this platform only provides point cloud given by depth camera. Other modalities such as RGB, normal, shading, etc. from multiple, configurable views should also be provided, to explore direct end-to-end training from raw sensor input, rather than known object target information like their 6D poses. This point should be an easy add-on though.
>
> Yes, I very much agree with you. Regarding visual input, we have also tried it. Isaac Gym can use RGBD cameras to provide us with visual information, which can be directly used as image input or processed into other modal data such as point clouds. But there is actually a problem is that the parallelism of Isaac Gym's cameras is not very good. It can only obtain images one by one env serially, which will greatly slow down the running speed. At the same time, the training of the dexterous hand is very difficult and greatly depends on the high sampling efficiency, so we do not use the input of other modalities. But this is undoubtedly very important, so we provide an example in our Github repo (see [here](https://github.com/PKU-MARL/DexterousHands/blob/main/bi-dexhands/tasks/shadow_hand_point_cloud.py)) for other modal inputs for future exploration or development.
>
> > Ease of task customization: I read through the paper and appendix but didn't find detailed descriptions on designing customized tasks. The authors only mention that objects from provided YCB and SAPIEN datasets could be used to replace the current basis, but how to generate a ground truth target pose or trajectory for every new object, or even a completely new task, is not given. This point is essential for users who want to develop based on this platform and create their own specific tasks.
>
> Yes, customizable tasks are very important to the development of a platform. Bi-DexHands not only support customizing the use of objects in the YCB and Sapien datasets to replace existing items (see [here](https://github.com/PKU-MARL/DexterousHands/blob/main/docs/customize-the-environment.md#How_to_replace_objects_in_existing_tasks_with_items_from_YCB_and_Sapien_datasets)), but also support custom tasks more conveniently. To facilitate this, we have written an additional markdown file to help users define their own tasks using Bi-DexHands, see [here](https://github.com/PKU-MARL/DexterousHands/blob/main/docs/customize-the-environment.md#How_to_create_your_own_task).
>
> > Different reward design for RL: In the appendix, each of 20 tasks is detailed with a given reward function with some hardcoded numbers. I wonder how are these numbers obtained by the authors and whether there are pre-experiments testing different types of rewards, because the reward function design is important to RL and the design details should be explored.
>
> Yes, designing a reward function is very important for an RL task. I would like to introduce the method of our reward design in detail. In general, our reward design is goal-based and follows the same set of logic. For object catch tasks, our reward is simply related to the difference between the pose of the object and the target. The closer the object is to the target, the greater the reward. For other tasks that require the hand to hold the object, our reward generally consists of three parts: the distance from the left hand to the target point on the object that the left-hand needs to operate, the distance from the right hand to the target point on the object that the right-hand needs to operate, and the distance from the object to the target.
> The principle of our design is to conform to human intuition based on completing the task and to make the reward function structure as unified as possible. This unified reward function structure is also one of the requirements of Meta RL and Multi-task RL environment design. However, because the scenarios of each task are different, the hyperparameters of the reward function will inevitably be different. We have tried our best to avoid manual reward shaping for each task provided that RL can be successfully trained.
>
> > The evaluation of some RL algorithms are not finished yet.
>
> Yes, you are right. Because we are a comprehensive RL benchmark, including RL, MARL, MTRL, MLRL, OffRL and so many tasks, it is very difficult to benchmark all algorithms. But we are working hard in this direction. At present, the TRPO, HATRPO algorithm has been newly evaluated, check our revision paper. But since TRPO-type algorithms need to solve the hessian matrix, which takes a long time, so we only completed the benchmark for part of the task. We will run through the rest of the time and continue to update during the rebuttal period.

---

> > ### Author Response · Authors · 2022-08-13
> > **Response to reviewer 5Yjg (2/2)**
> >
> > > The paper is well written overall. I recommend a proof-reading on grammars and put more essential details about the dataset itself to the main paper from supplementary materials (a summary of A.1 and A.2)
> >
> > Thanks for your suggestion. We have added more clarifications to the Appendix for a better understanding of our work.
> >
> > > Yes, this work is novel and different from previous works. I recommend the authors considering more diverse tasks and algorithms mentioned in these previous works:
> > https://ieeexplore.ieee.org/stamp/stamp.jsp?arnumber=8989777 https://ieeexplore.ieee.org/stamp/stamp.jsp?arnumber=9196527 https://ieeexplore.ieee.org/stamp/stamp.jsp?arnumber=9035030 https://arxiv.org/pdf/1904.11382.pdf https://arxiv.org/pdf/2010.08587.pdf
> >
> > Yes, we will consider adding more diverse tasks to bi-dexhands next to continue promoting the development of bimanual dexterous manipulation. At present, there is relatively little related work on dual dexterous hands, and we are also considering how to design the task, so your suggestion is very useful, thank you!
> >
> > > 1, How good is the physical contact rich actions simulated in the proposed platform, or Issac Gym?
> > > 2, How hard is it to import other dexterous manipulators and design new tasks to the workspace?
> > > 3, How to further evaluate the trained RL agents in real world?
> >
> > Thank you for your interest in further extending our work! I am very happy to communicate with you in this regard.
> > For 1, as far as I know, the physical simulation effect of PhysX used by Isaac Gym is very good, especially in contact rich. In fact that's one of the reasons why we chose Isaac Gym as my platform. In [this paper](https://github.com/AndrejOrsula/master_thesis/blob/master/master_thesis.pdf) page 30, the author introduce and compare the physics engine of the mainstream RL simulation environment. It can be found that Isaac Gym is one of the best ones. You can also take a look if you are interested.
> > For 2, in fact, if you are familiar with Isaac Gym, it is easy to import other dexterous manipulators and design new tasks. Personally, I think Isaac Gym's API is more user-friendly and modern than mujuco or pybullet. The disadvantage may be that Isaac Gym is still in the development stage, and there will inevitably be some compatibility bugs, but in general it does not affect his use. Therefore, I highly recommend isaacgym as a simulation environment, because its high sampling efficiency is very important for RL. If you have any questions or difficulties about importing other dexterous manipulators and design new tasks, see our Github example [customize-the-environment.md](https://github.com/PKU-MARL/DexterousHands/blob/main/docs/customize-the-environment.md#How_to_create_your_own_task) and [Change-the-type-of-dexterous-hand.md](https://github.com/PKU-MARL/DexterousHands/blob/main/docs/Change-the-type-of-dexterous-hand.md), or feel free to contact me in an issue!
> > For 3, how to use the model trained by simulation in real world, called sim2real, is a very important problem in robotics. Regarding how dexterous hands can achieve sim2real, you can refer to OpenAi previous work on [here](https://openai.com/blog/learning-dexterity/). As we mentioned in futurework, we are also working in this direction and hope to transfer our trained models to real world.

---

> > > ### Comment · Reviewer_5Yjg · 2022-08-18
> > > **Response to author's answer**
> > >
> > > I am good with the author's answers to my concerns in data modality, task customization, and simulation platform. It's good to see it can support new objects and customized tasks. However, the parallel implementation of multiple sensors and modality remains a bottleneck for this benchmark.

---

> > > > ### Author Response · Authors · 2022-08-26
> > > > **Thank you for your response**
> > > >
> > > > Thank you very much for your response. We are also recently exploring how to achieve efficient parallelization of multiple sensors and hope to implement it one day. We will ensure long-term maintenance of Bi-DexHands and put in more effort to further improve it.

---

### Official Review · Reviewer_EVVH · 2022-07-28
**A benchmark for bimanual dexterous hand manipulation**

**Rating:** 7
**Confidence:** 5
**Correctness:** Yes

**Strengths:**

- Though dexterous hand manipulation is an important topic and draws much attention, there are no good benchmarks for dexterous hand manipulation in the bimanual setting. This benchmark provides a good test bed for bimanual dexterous hand manipulation.

- Benchmarked many different categories of  RL methods, including single-agent RL, Multi-agent RL (MARL), Offline RL, Multi-task RL, and Meta-RL.

- The simulation speed is decent, though it is mainly because of Issac Gym.

- The documentation is good overall, though there are still some minor issues.

**Weaknesses:**

- Simulation

  - According to the figures shown in the Github repo, I feel there might be some simulation issues presented in this benchmark, e.g.,

    - In [this figure](https://github.com/PKU-MARL/DexterousHands/blob/main/assets/image_folder/scissors.gif), the scissors start to move before being touched.

    - In [this figure](https://github.com/PKU-MARL/DexterousHands/blob/main/assets/image_folder/pen.gif?raw=true), the movement of pen cap seems incorrect.

    - As far as I know, the [simulation of opening the cap](https://github.com/PKU-MARL/DexterousHands/blob/main/assets/image_folder/bottle_capv2.gif) of bottle is non-trivial. Can you provide more details about how you implement this? And where are the object models come from?

  - Simulating everything correctly is tricky, but this is somehow very important in build simulated robotic tasks, especially when using RL agent (which always exploits the bug in the simulator)

- Performance metrics

  - Currently, the performance on the tasks is measured by average rewards, but this is not very intuitive. For example, people cannot easily understand whether an agent with ~100 reward is good or bad, without looking into the details of this task.

  - Therefore, I hope there can be more intuitive ways to measure the performance. One possible solution can be using some binary-based metrics like success rates.

- Observations of the tasks

  - The observations are state-based (a vector describing the states of everything in the environment). This is good to kick off the research, but not a realistic setting. In real world, agent always needs to estimate the states of other objects by visual observations. It would be better to consider incorporating visual observations in these tasks.

- Benchmarked methods

  - Only RL-based methods are benchmarked. However, other methods like motion planning, model predictive control (MPC) are also widely used in robotics research. If this benchmark is not purely designed for RL, then other types of methods should also be benchmarked.


**Additional Feedback:**

None

**Clarity:**

The overall structure of the paper is clear and easy to follow.



Something I want the authors to clarify:

- What controllers are used to control the dex hands? And how do you determine what controllers to use? I guess something like PID controllers are used.


**Documentation:**

The documentation is overall clear, but lack of some details.

- In the [environment document](https://github.com/PKU-MARL/DexterousHands/blob/main/docs/environments.md), only 5 tasks (ShadowHandOver    ShadowHandCatchUnderarm    ShadowHandTwoCatchUnderarm    ShadowHandCatchAbreast    ShadowHandOver2Underarm) are explained in details.

- The tasks shown in [here](https://github.com/PKU-MARL/DexterousHands#tasks) (16 tasks), [here](https://github.com/PKU-MARL/DexterousHands#figures) (20 tasks),  Table 1 in the main paper (13 tasks), and appendix A2 (20 tasks) are not aligned. So which one shows the full set of tasks?


**Relation To Prior Work:**

Yes.

However, I did not find an explanation about how the objects from YCB and SAPIEN are used. For example, which task uses YCB objects and which task uses SAPIEN objects? How are the objects selected?

**Summary And Contributions:**

This submission proposes Bimanual Dexterous Hands Benchmark (Bi-DexHands), a simulator that involves two dexterous hands with tens of bi-manual manipulation tasks and thousands of target objects (mainly come from YCB and SAPIEN).

The authors claim Bi-DexHands has five key features: Isaac Gym Efficiency, Comprehensive RL Benchmark, Heterogeneous-agent Cooperation, Task Generalization, and Cognition.

To the best of my knowledge, Bi-DexHands is the first simulation environment which focus on bimanual dexterous hand manipulation tasks. This benchmark will probably attract researchers from robotics and reinforcement learning.

---

> ### Author Response · Authors · 2022-08-13
> **Response to reviewer EVVH (1/2)**
>
> We would like to sincerely thank you for reviewing our paper and providing constructive comments!
>
> Here we address your comments below:
>
> > In this figure, the scissors start to move before being touched.
>
> This is a bug that may only appear in the test before, we have fixed it, see [here](https://github.com/PKU-MARL/DexterousHands/blob/main/assets/image_folder/scissors.gif).
>
> > In this figure, the movement of pen cap seems incorrect.
>
> Our pen model is from Sapien datasets. The pen and pen cap are connected by a translation joint, which may be different from the real pen and pen cap.
>
> > As far as I know, the simulation of opening the cap of bottle is non-trivial. Can you provide more details about how you implement this? And where are the object models come from?
>
> Yes, the simulation of opening the cap of bottle is non-trivial and is still difficult to implement. Like the pen and pen cap, most of our object models are from Sapien datasets. The bottle and bottle cap are connected by a translational joint, so our setting will be similar to opening the pen cap, putting them in the same baby age and level. However, the new version of Isaac Gym now provides demos similar to screwing (see [here](https://sites.google.com/nvidia.com/factory/)) and may provide models to implement similar tasks. It has been added to our Todo list, see [here](https://github.com/PKU-MARL/DexterousHands#future-plan).
>
> > Simulating everything correctly is tricky, but this is somehow very important in build simulated robotic tasks, especially when using RL agent (which always exploits the bug in the simulator)
>
> Yes, we agree with that too, so we try to keep the simulation as realistic as possible when designing the environment. For example, using Isaac Gym, the PhysX physics engine it uses is more realistic. Use a high-resolution vhacd algorithm when loading objects to ensure that the collision mesh is as fine as possible.
>
> > Currently, the performance on the tasks is measured by average rewards, ... using some binary-based metrics like success rates.
>
> Yes, using average reward is not very intuitive. But there is a problem that some of our tasks are still difficult to complete, providing to the community as a challenge. So using the success rate directly may not be a good measure of these difficult tasks. Therefore, we are considering using the reward after normalization as a performance indicator. It has been added to our Todo list, see [here](https://github.com/PKU-MARL/DexterousHands#future-plan).
>
> > The observations are state-based (a vector describing the states of everything in the environment). ... visual observations in these tasks.
>
> Regarding the selection of our observation, we are providing everything in the environment about hands and objects to everyone. The original intention is to allow everyone to choose the observation according to their needs.
> Regarding visual input, we have also tried it. Isaac Gym can use RGBD cameras to provide us with visual information, which can be directly used as image input or processed into a point cloud. We provide an example for point cloud observation in our Github repo for reference (see [here](https://github.com/PKU-MARL/DexterousHands/blob/main/bi-dexhands/tasks/shadow_hand_point_cloud.py)).
>
> > Only RL-based methods are benchmarked. However, other methods like motion planning, model predictive control (MPC) are also widely used in robotics research. If this benchmark is not purely designed for RL, then other types of methods should also be benchmarked.
>
> Yes, methods other than RL are also very important. While our repo is currently focusing on a broad range of RL algorithms, we would also be more than happy to benchmark other control methods that are widely used in robotics research.
>
> > What controllers are used to control the dex hands? And how do you determine what controllers to use? I guess something like PID controllers are used.
>
> Yes, the choice of controller is very important. We are using the PD controller, the method that comes with Isaac Gym. This setting was decided by referring to the IsaacGymEnvs shadow hand environment, which is enough for us at present. Considering the possibility of sim2real in the future, it is also a very important feature to develop other more powerful controllers like PID.

---

> > ### Author Response · Authors · 2022-08-13
> > **Response to reviewer EVVH (2/2)**
> >
> > > However, I did not find an explanation about how the objects from YCB and SAPIEN are used. For example, which task uses YCB objects and which task uses SAPIEN objects? How are the objects selected?
> >
> > The objects in the YCB dataset are used for our object-catching tasks. Because our object-catching environment is only related to the pose of the object, we can arbitrarily replace objects of suitable size in the YCB dataset. Other tasks use items from the Sapien dataset, and can also use other objects from the same category in Sapien dataset. However, because it is related to the shape of the object, some additional operations are required. We have added examples to Github to show how to use objects from YCB and Sapien datasets, see [here](https://github.com/PKU-MARL/DexterousHands/blob/main/docs/customize-the-environment.md).
> >
> > > In the environment document, only 5 tasks (ShadowHandOver ShadowHandCatchUnderarm ShadowHandTwoCatchUnderarm ShadowHandCatchAbreast ShadowHandOver2Underarm) are explained in details.
> >
> > Thank you for reminding. We have updated the introduction of all tasks in the document, see [here](https://github.com/PKU-MARL/DexterousHands/blob/main/docs/environments.md).
> >
> > > The tasks shown in here (16 tasks), here (20 tasks), Table 1 in the main paper (13 tasks), and appendix A2 (20 tasks) are not aligned. So which one shows the full set of tasks?
> >
> > To be precise, we have 20 different tasks. The 16 tasks here are because there are still some that have not been updated in time. We have updated the introduction of all 20 tasks. The 13 tasks here are the stages of infant motor development that we referenced when designing the tasks. There are some tasks of the same type that belong to the same stage, so we merge them together. For example, opening and closing the door, we only use one column to introduce, but there are actually 4 different tasks: **DoorOpenInward**, **DoorCloseInward**, **DoorCloseOutward**, **DoorOpenOutward**; we only use one column for object-catching task, and there are actually 5 tasks.

---

> > > ### Comment · Reviewer_EVVH · 2022-08-14
> > > **Response**
> > >
> > > Thank the authors for the detailed response.
> > >
> > > About the bottle cap task:
> > >
> > > I understand that correctly simulating a realistic bottle cap is very hard, so I think it is ok to make some simplifications. However, if the bottle and bottle cap are connected by a prismatic joint, then I would suggest renaming this task to remind the users that this is totally different from the realistic bottle cap opening.
> > >
> > > About the performance metric:
> > >
> > > The normalized reward is better than the unnormalized counterpart, but I still believe defining a clear success metric would be better. And I am also wondering if some of your tasks are still difficult to complete, how do you verify those tasks are solvable? Did you use anything like motion planning or manual control?
> > >
> > > About the visual observations:
> > >
> > > I am glad that you provide the point cloud observations. However, have you tried to solve the tasks based on the visual observations? Providing visual observations is a little tricky because you need to tune the parameters of the cameras (intrinsic, extrinsic, resolutions, etc.). Therefore, verification is pretty crucial for visual observation.
> > >
> > > About the controllers:
> > >
> > > Do you use velocity PD or position PD?

---

> > > > ### Author Response · Authors · 2022-08-16
> > > > **Further response to reviewer EVVH**
> > > >
> > > > We deeply appreciate your precious comments. As for the problems you raised, we would like to address them as follows.
> > > >
> > > > > About the bottle cap task:
> > > > I understand that correctly simulating a realistic bottle cap is very hard, so I think it is ok to make some simplifications. However, if the bottle and bottle cap are connected by a prismatic joint, then I would suggest renaming this task to remind the users that this is totally different from the realistic bottle cap opening.
> > > >
> > > > You are right, We will further indicate that this bottle does not need to be twisted where the name appears to remind the users. Check our revision paper.
> > > >
> > > > > About the performance metric:
> > > > The normalized reward is better than the unnormalized counterpart, but I still believe defining a clear success metric would be better.
> > > >
> > > > Yes, we also think that the success matrix is the most suitable for our goal-based task, and we have added it to our Todo list, see [here](https://github.com/PKU-MARL/DexterousHands#future-plan).
> > > >
> > > > > And I am also wondering if some of your tasks are still difficult to complete, how do you verify those tasks are solvable? Did you use anything like motion planning or manual control?
> > > >
> > > > Thank you for your concern. The solvability of our environment is our primary consideration during development. In fact, only 2 of the 20 tasks are currently unfinish -- TwoCatch Underarm and Pour Water. TwoCatch Underarm will clearly see signs of completion during the training process: it even completes the task sometimes. Just due to instability, it chose to throw only one object in a more stable way in later stage of training. But obviously, this task is solvable, as long as it can learn to throw the object successively or throw the object with different posture. Pour water can only learn to grasp the handle of the kettle, so we choose manual control to verify the solvability. Manually set the spatial pose of the hand after grabbing the kettle to ensure the task is solvable.
> > > >
> > > > > About the visual observations:
> > > > I am glad that you provide the point cloud observations. However, have you tried to solve the tasks based on the visual observations? Providing visual observations is a little tricky because you need to tune the parameters of the cameras (intrinsic, extrinsic, resolutions, etc.). Therefore, verification is pretty crucial for visual observation.
> > > >
> > > > Very appreciated question! We often train RL with visual input in Isaac Gym and have tried it in Bi-DexHands. But the problem is that the parallelism of Isaac Gym's cameras is not very good. It can only obtain images one by one env serially, which will greatly slow down the running speed. At the same time, the training of the dexterous hand is very difficult and greatly depends on the high sampling efficiency, so we do not use the input of other modalities. Below we will do a simple experiment to illustrate it.
> > > >
> > > > Regarding camera parameters, we adjust extrinsic and resolutions with camera image visualization to suit our task. [Here](https://github.com/PKU-MARL/DexterousHands/blob/main/assets/image_folder/point_cloud/point_cloud_image.png) is an example of ShadowHandOver. For intrinsic, Some advanced uses such as deprojecting depth images to 3D point clouds require complete knowledge of the projection terms used to create the output images. To aid in this, Isaac Gym provides access to the projection and view matrix used to render a camera's view. For specific operations, see the documentation we mentioned below.
> > > >
> > > > To better answer your question, we do a simple experiment and result [here](https://github.com/PKU-MARL/DexterousHands/blob/main/assets/image_folder/point_cloud/point_cloud.png). We replace the object information with point clouds in the case of a small number of environments, and use PointNet to extract point cloud features. It can be seen that under the same episode and same number of environments, the performance of point cloud input is not as good as full state input, but it can also achieve some performance. But also using an RTX 3090 GPU, the point cloud RL has only 200+ fps, and the full state can reach 30000+. In fact, we can only open up to 256 environments when using point clouds, and train the whole thing. This was a problem with Isaac Gym's poor parallel support for cameras, so we didn't use point clouds or other visual inputs as our baselines, but they could.
> > > >
> > > > Thank you for the appreciative review that makes our paper and Github repo even better! We have organized the detailed information and examples about visual input into documents and put them on Github, see [here](https://github.com/PKU-MARL/DexterousHands/blob/main/docs/point-cloud.md).
> > > >
> > > > > About the controllers:
> > > > Do you use velocity PD or position PD?
> > > >
> > > > Thank you for your concern, we are using position PD currently.

---

### Official Review · Reviewer_BDzh · 2022-07-28
**Good environments and tasks to benchmark existing RL algorithms but lacks some technical details.**

**Rating:** 7
**Confidence:** 5

**Strengths:**

* easy to use environments.
* good efficiency
* pretty successful attempt to create a benchmark in which you can compare ai with a human skills which might have positive impact in future general ai research.
* Provided  relationship between their dexterous tasks and the 70 motor skills of humans at different ages.

**Weaknesses:**

* Didn't include observation space design for different tasks. It is impossible to understand what is the difference in observation between MARL and Single Agent versions without looking into the source code.
* Same issue with reward. Reward function was not  described in paper. Rewards in different even similar tasks have very different scale. Didn't provide maximum reward for each task. For example in 'Two Catch Underarm' algorithms achieved only one catch. Looks like to solve it you need to get 27+ reward.
* No analysis of the different algorithm  performance.
* for some envs 100m of the steps was not enough.

**Additional Feedback:**

Overall pretty good benchmark. But you can add more information.
I'd like to see in the paper:
1) observation, reward and action configuration for different tasks. Or at least demonstrate how you build it for one example task.
2) Maybe normalize reward so it will have same scale for different tasks.
3) It would be nice to add support for the default IsaacGymEnvs RL library rl-games.
4) Explain benchmark results. Why PPO is the best and SAC cannot even solve it.

**Clarity:**

The paper is well written and easy to follow. The visuals and tables significantly help my understanding of the paper.
But not enough explanation about how reward is calculated. Also environments were designed in the way that on some of them we see reward ~1000 and ~20 on others. I think this part could be unified.

**Correctness:**

Benchmark was done using a couple of on-policy MARL algorithms and one off-policy SAC which is the SOTA for a lot of continuous action space tasks. But in this case SAC didn't work well which is consistent with IsaacGym ShadowHand environment but requires additional investigation in future.

**Documentation:**

The author provides an open-source link to GitHub with good readme file where we can find almost all needed information about environments. It lacks more details about reward and observation space design for each task.

**Ethics:**

I don't see any significant ethics concerns.

**Relation To Prior Work:**

One of the most related prior work which  is the IsaacGymEnvs shadow hand environment which worked as a baseline for current envs.
Also they mentioned OpenAI's Rubik's Cube. But this work's goals are much different from prior works and it doesn't make sense to compare them a lot.

**Summary And Contributions:**

* They chose IsaacGym. In my opinion it is currently the best library for the robotics or physics research. It makes possible to run one experiment in 30 minutes instead of the days if use Mujoco instead.
* First released benchmark for bimanual dexterous hands manipulation with such a good performance.
* Added comparison between environments and human skills at different age. Which demonstrated that current skills are pretty basic for
 human and achievable by baby which cannot be said about computer.
* There were introduced 20 different tasks and tested on the most popular RL algorithms including Meta RL and Offline RL. Surprisingly SAC didn't work well. Which may require additional investigation.

---

> ### Author Response · Authors · 2022-08-13
> **Response to reviewer BDzh**
>
> We would like to sincerely thank you for reviewing our paper and providing constructive comments! We appreciate your enthusiasm for the work!
>
> Here we address your comments below:
>
> > Didn't include observation space design ... looking into the source code.
>
> For the design of observation, action, and reward, we have a detailed description in the Appendix A.2. Recently, we have also made related instructions in our Github, see [here](https://github.com/PKU-MARL/DexterousHands/blob/main/docs/environments.md).
>
> > Same issue with reward. Reward .... need to get 27+ reward.
>
> Yes, designing a reward function is very important for an RL task. I would like to introduce the method of our reward design in detail. In general, our reward design is goal-based and follows the same set of logic. For object catch tasks, our reward is simply related to the difference between the pose of the object and the target. The closer the object is to the target, the greater the reward. For other tasks that require the hand to hold the object, our reward generally consists of three parts: the distance from the left hand to the target point on the object that the left-hand needs to operate, the distance from the right hand to the target point on the object that the right-hand needs to operate, and the distance from the object to the target.
> The principle of our design is to conform to human intuition based on completing the task and to make the reward function structure as unified as possible. This unified reward function structure is also one of the requirements of Meta RL and Multi-task RL environment design. However, because the scenarios of each task are different, the hyperparameters of the reward function will inevitably be different. We have tried our best to avoid manual reward shaping for each task provided that RL can be successfully trained.
>
> > Benchmark was done using a couple of ... but requires additional investigation in future.
>
> Regarding why SAC can not work well, we have conducted some experiments and explanations in the Appendix C. Overall, we believe that this is indeed a phenomenon worthy of further study in the future.
>
> > The author provides an open-source link to ... space design for each task.
>
> Thank you for your reminder. We have added an additional file to the readme for a detailed description, check [here](https://github.com/PKU-MARL/DexterousHands/blob/main/docs/environments.md).
>
> > Overall pretty good benchmark. But you can add more information. I'd like to see in the paper:
> 1, observation, reward and action configuration for different tasks. Or at least demonstrate how you build it for one example task.
> 2, Maybe normalize reward so it will have same scale for different tasks.
> 3, It would be nice to add support for the default IsaacGymEnvs RL library rl-games.
> 4, Explain benchmark results. Why PPO is the best and SAC cannot even solve it.
>
> For 1, we have added detailed observation, reward, and action configuration for different tasks in both Github and Appendix, see [here](https://github.com/PKU-MARL/DexterousHands/blob/main/docs/environments.md).
> For 2 and 3, yes, I think the reward for normalizing is meaningful. In fact, we do rough normalization when doing meta and multi-task training. These two points have been added to our Todo list, see [here](https://github.com/PKU-MARL/DexterousHands#future-plan).
> For 4, we have carried out some experiments and explanations in the Appendix C. This is also a very surprising place for us. We have conducted a lot of experiments and double-checks, and the conclusion is mentioned in the Appendix. We also hope to investigate this issue with the community, and maybe there will be some surprising findings.

---

### Author Response · Authors · 2022-08-13
**To all reviewers: thank you for your comments and feedback!**

We thanks everyone's comments to help our paper better! Based on the feedback from reviewers, we were able to identify a number of areas for improvement. We have updated the revision paper, code, and document after reading to the suggestions of all reviewers. At the same time, many new features have been added to our benchmark. Here we make an updated summary for all reviewers:

- The revision paper and appendix are updated, and the modifications are marked with highlight.
- We add a detailed introduction to the environment and put it on GitHub as a markdown file: [https://github.com/PKU-MARL/DexterousHands/blob/main/docs/environments.md](https://github.com/PKU-MARL/DexterousHands/blob/main/docs/environments.md)
- We add a example to Github to show how to use objects from YCB and Sapien datasets and How to create your own task, see [https://github.com/PKU-MARL/DexterousHands/blob/main/docs/customize-the-environment.md](https://github.com/PKU-MARL/DexterousHands/blob/main/docs/customize-the-environment.md).
- We add a new feature for changing different types of dexterous hands, see GitHub for more details: [https://github.com/PKU-MARL/DexterousHands/blob/main/docs/Change-the-type-of-dexterous-hand.md](https://github.com/PKU-MARL/DexterousHands/blob/main/docs/Change-the-type-of-dexterous-hand.md)
- We add a new feature for adding a robotic arm to the base of the dexterous hand. For more details, see GitHub: [https://github.com/PKU-MARL/DexterousHands/blob/main/docs/Add-a-robotic-arm-drive-to-the-dexterous-hand.md](https://github.com/PKU-MARL/DexterousHands/blob/main/docs/Add-a-robotic-arm-drive-to-the-dexterous-hand.md)
- We followed the reviewer's useful advice on the direction of bi-dexhands and added a todo list as a future direction: [https://github.com/PKU-MARL/DexterousHands#Future-Plan](https://github.com/PKU-MARL/DexterousHands#Future-Plan)
- We have been benchmarking other remaining RL algorithms, and added their curves to the result figure (Figure. 3) of the revision paper. At present, the TRPO, HATRPO algorithm has been newly evaluated. But since TRPO-type algorithms need to solve the hessian matrix, which takes a long time, so we only completed the benchmark for part of the task. We will run through the rest of the time and continue to update during the rebuttal period.

---

> ### Author Response · Authors · 2022-08-19
> **To all reviewers: We added some baselines and a detailed description of visual input RL**
>
> Thank you for your valuable suggestions on our work, making our paper better. During the rebuttal period, we added some baselines and made a simple experimental description of the visual input RL on Bi-DexHands. We hope these new additions will make our paper better. Here we make an updated summary for all reviewers:
>
>  - We newly added TRPO, HATRPO baselines on 20 tasks, see our revision paper (Fig. 3).
>  - We newly added documentation and experiments for point cloud RL and related visual inputs, see [here](https://github.com/PKU-MARL/DexterousHands/blob/main/docs/point-cloud.md).

---

### Meta-Review · Area_Chair_Uxsk · 2022-09-02

**Recommendation:** Accept
**Confidence:** 4

**Metareview:**

The paper provides a variety of environment simulators and settings involving bimanual dexterous manipulation. These settings encompass many popular paradigms in RL today, including offline RL and multi-agent RL. I am glad to see that the reviewer discussion was fruitful and led to an improved submission. I would further encourage the authors to include more results with visual observations (as suggested by Reviewer EVVH). Some sort of visual input is becoming increasingly common in real-world RL-trained robots, and so benchmarks for this setting are important.

---

### Decision · Program_Chairs · 2022-09-16

Accept